# Histone Deacetylase Inhibitors as a Therapeutic Strategy to Eliminate Neoplastic “Stromal” Cells from Giant Cell Tumors of Bone

**DOI:** 10.3390/cancers14194708

**Published:** 2022-09-27

**Authors:** Sanne Venneker, Robin van Eenige, Alwine B. Kruisselbrink, Ieva Palubeckaitė, Alice E. Taliento, Inge H. Briaire-de Bruijn, Pancras C. W. Hogendoorn, Michiel A. J. van de Sande, Hans Gelderblom, Hailiang Mei, Judith V. M. G. Bovée, Karoly Szuhai

**Affiliations:** 1Department of Pathology, Leiden University Medical Center, 2333 ZA Leiden, The Netherlands; 2Department of Cell and Chemical Biology, Leiden University Medical Center, 2333 ZC Leiden, The Netherlands; 3Department of Orthopedic Surgery, Leiden University Medical Center, 2333 ZA Leiden, The Netherlands; 4Department of Medical Oncology, Leiden University Medical Center, 2333 ZA Leiden, The Netherlands; 5Sequencing Analysis Support Core (SASC), Leiden University Medical Center, 2333 ZC Leiden, The Netherlands

**Keywords:** bone neoplasm, giant cell tumor of bone, H3F3A, histone H3.3 variants, epigenetics, histone deacetylase, romidepsin, panobinostat

## Abstract

**Simple Summary:**

Giant cell tumor of bone (GCTB) is an intermediate bone neoplasm which consists of several cell populations, including the neoplastic “stromal” cells. These cells harbor a mutation in one of the histone H3.3 genes (H3F3A), and are therefore considered as the driving component of GCTB. This mutation causes changes in the epigenetic landscape, leading to aberrant gene expression patterns that may drive tumor growth. Surgery is currently the only curative treatment option because contemporary systemic therapies cannot remove the neoplastic cells from GCTB lesions, leading to re-outgrowth of the tumor when the treatment is discontinued. Therefore, the aim of this study was to explore whether therapeutic targeting of the epigenome can eliminate the neoplastic cells from GCTB lesions. The findings from this study indicate that histone deacetylase (HDAC) inhibitors may represent such a treatment strategy, which could improve the quality of life of GCTB patients who currently require life-long treatment.

**Abstract:**

The neoplastic “stromal” cells in giant cell tumor of bone (GCTB) harbor a mutation in the *H3F3A* gene, which causes alterations in the epigenome. Current systemic targeted therapies, such as denosumab, do not affect the neoplastic cells, resulting in relapse upon treatment discontinuation. Therefore, this study examined whether targeting the epigenome could eliminate the neoplastic cells from GCTB. We established four novel cell lines of neoplastic “stromal” cells that expressed the H3F3A p.G34W mutation. These cell lines were used to perform an epigenetics compound screen (*n* = 128), which identified histone deacetylase (HDAC) inhibitors as key epigenetic regulators in the neoplastic cells. Transcriptome analysis revealed that the neoplastic cells expressed all HDAC isoforms, except for HDAC4. Therefore, five HDAC inhibitors targeting different HDAC subtypes were selected for further studies. All GCTB cell lines were very sensitive to HDAC inhibition in both 2D and 3D in vitro models, and inductions in histone acetylation, as well as apoptosis, were observed. Thus, HDAC inhibition may represent a promising therapeutic strategy to eliminate the neoplastic cells from GCTB lesions, which remains the paramount objective for GCTB patients who require life-long treatment with denosumab.

## 1. Introduction

Giant cell tumor of bone (GCTB) is defined as an intermediate bone neoplasm, which indicates that it is a locally aggressive but rarely metastasizing tumor [1]. GCTB usually originates at the end of long bones of skeletally mature young adults (20 to 45 years old) and has a local recurrence rate of 15–50% [1,2]. Despite the high recurrence rate, fewer than 1% of all GCTB cases undergo malignant transformation due to an as-yet not completely identified mechanism [3]; the typically slow-growing pulmonary metastases are rarely observed [1].

Histologically, GCTB is characterized by three different cell populations: mononuclear spindle-shaped “stromal” cells, mononuclear macrophage-like osteoclast precursor cells, and multinucleated osteoclast-like giant cells [4]. The neoplastic “stromal” cells secrete high levels of various chemokines (e.g., CCL2 and CXCL12) to attract osteoclast precursor cells from the circulation to the tumor site [4,5]. The attracted monocytes will upregulate the expression of the RANK receptor (TNFRSF11A) due to the additional secretion of CSF1 (M-CSF1) by the neoplastic cells [6]. Furthermore, the neoplastic cells produce high levels of RANK ligand (TNFSF11), which will induce monocyte differentiation, fusion, and ultimately the formation of the characteristic multinucleated giant cells [7]. These formed giant cells exhibit osteoclast-like bone resorption features, causing the characteristic osteolytic phenotype of GCTB.

GCTB frequently harbors a heterozygous hotspot mutation in one of the genes encoding for histone H3.3 (*H3F3A*) (92% of the cases), and this mutation is exclusively observed in the neoplastic “stromal” cells [8]. The affected glycine residue (p.G34W and, to a lesser extent, p.G34L mutations) is highly specific for GCTB, and helps to discriminate GCTB from chondroblastoma (characterized by p.K36M mutations in *H3F3A* and *H3F3B*) and other giant cell-rich tumor types [9,10]. The identification of the *H3F3A* variant has not only led to the development of novel diagnostic tools, but also confirmed that the “stromal” cells are the driving neoplastic component of GCTB, because these cells can be maintained in vitro and cause osteolytic bone lesions in orthotopic mouse models [11,12,13]. Moreover, knockdown or correction of the *H3F3A p.G34W* gene leads to a reduction in cell proliferation, cell migration, colony formation, and RANK ligand production in vitro, and inhibits tumor formation in mice [12,13].

Although the oncogenic mechanisms responsible for the neoplastic outgrowth of the tumor cells are rather complex [14] and relate to an interplay between the *H3F3A* mutation and telomere regulation in the neoplastic cells, the exact targets for oncogenesis, and thus, potential targets for therapy, have not yet been identified. Due to the incorporation of the H3F3A p.G34W oncohistone into the chromatin, several histone modifications (e.g., H3K36me3 and H3K27me3) are globally reduced or redistributed [13,15,16,17]. As these histone marks are changed beyond H3F3A p.G34W incorporation sites, the epigenetic landscape of the neoplastic cells is highly altered, which causes aberrant gene transcription patterns which may drive the formation of GCTB. The current understanding of the molecular mechanism behind H3F3A p.G34W-induced remodeling of the epigenome focuses around impaired osteogenic or myogenic differentiation of the neoplastic cells, resulting in the pathogenic secretion of ligands that recruit and activate osteoclast-like cells [13,16,18].

The standard of curative care for GCTB patients is surgery. If tumors are regarded unresectable or to facilitate surgical treatment, patients can be treated with systemic targeted therapy (i.e., denosumab or bisphosphonates) [2,19,20,21]. Denosumab, a fully human antibody against the RANK ligand, inhibits the formation of giant cells and thus stops the giant cell-driven bone resorption process. However, life-long treatment, when irresectable, may be required, because the neoplastic cells are not targeted by denosumab [22], resulting in relapse upon treatment discontinuation. Therefore, there is a need to develop novel therapeutic strategies which directly target the neoplastic cells in GCTB.

Due to the presence of a pathognomonic histone mutation (i.e., *H3F3A p.G34W*) in GCTB, in the current study we examined whether targeting the epigenetic landscape could eliminate the neoplastic cells from GCTB lesions. To achieve this, four novel cell lines of neoplastic cells from GCTB patients, which were pre-operatively treated with denosumab, were established. These four cell lines expressed the *H3F3A p.G34W* mutation in vitro and were used to perform a broad epigenetics compound screen (*n* = 128) to identify key epigenetic regulators in the neoplastic cells.

## 2. Materials and Methods

### 2.1. Cell Culture

Cell lines derived from the neoplastic “stromal” cells (L4040, L5077, L5345, and L5862) were generated at the Leiden University Medical Center (LUMC) from resected giant cell tumors of bone which were pre-operatively treated with denosumab (Figure 1 and Appendix A). All samples were handled and coded according to the “Code for Proper Secondary Use of Human Tissue in The Netherlands” (Dutch Federation of Medical Scientific Societies), and their use was approved by the LUMC ethical committee (B20.012). Cells were cultured in IMDM (Gibco, Invitrogen Life-Technologies, Scotland, UK) supplemented with 10% heat-inactivated fetal bovine serum (FBS) (F7524, Sigma-Aldrich, Saint Louis, MO, USA) and 1% insulin–transferrin–selenium (ITS-G) (Gibco). Cultures were maintained in a humidified incubator with 5% CO_2_ at 37 °C. A PCR-based mycoplasma test and short tandem repeat (STR) profiling (GenePrint 10 System, Promega, Madison, WI, USA) were performed monthly. Due to the time of establishment or slow growth rates, the L5077, L5345, and L5862 cell lines could not be included in all performed experiments (Appendix A).

### 2.2. Sanger Sequencing of H3F3A p.G34W

DNA was isolated from cultured cells using the High Pure PCR Template Preparation Kit (Roche, Basel, Switzerland) according to the manufacturer’s protocol. DNA (10–100 ng) was amplified with iQ™ SYBR^®^ Green Supermix (Bio-Rad Laboratories, Hercules, CA, USA) and *H3F3A* primers (FW: GACGTAATCTTCACCCTTTCAA and RV: TCCAGGTAAGATTATGGCTTCA) using a CFX96 Touch™ Real-Time PCR Detection System (Bio-Rad). Sanger sequencing was performed at Macrogen (Amsterdam, the Netherlands) using an ABI 3730xl system (Applied Biosystems™, ThermoFisher Scientific, Waltham, MA, USA).

### 2.3. H3F3A p.G34W Immunohistochemistry

Cultured cells were fixed in 4% formaldehyde (Q Path, VWR Chemicals, Radnor, PA, USA) and prepared for paraffin embedding with the Epredia™ Cytoblock™ Cell Block Preparation System (ThermoFisher Scientific). Sections were stained with hematoxylin and eosin (H&E) using the Leica ST5020-CV5030 Stainer Integrated Workstation (Leica Biosystems, Amsterdam, The Netherlands). The H3F3A p.G34W immunohistochemical stain (RM263, RevMAb, Biosciences, San Francisco, CA, USA) was performed with the Dako Omnis immunostainer (Agilent, Santa Clara, CA, USA) according to the standard diagnostic procedure at LUMC.

### 2.4. Compounds

A detailed list of all compounds (*n* = 128) included in the epigenetics compound library (L1900, Selleckchem, Houston, TX, USA) is presented in the Appendix A. The histone deacetylase (HDAC) inhibitors romidepsin (S3020, Selleckchem), panobinostat (S1030, Selleckchem), dacinostat (S1095, Selleckchem), quisinostat (S1096, Selleckchem), and fimepinostat (S2759, Selleckchem) were dissolved in DMSO.

### 2.5. Epigenetics Compound Screens

L4040 (15,000/well) and L5345 (15,000/well) cells were seeded in 96-well plates and allowed to attach overnight. Subsequently, cells were treated with the epigenetics compound library (Appendix A) at a final concentration of 1 µM. After 72 h of treatment, cells were fixed with 4% formaldehyde and stained with 2 µg/mL Hoechst 33342 (H1399, Invitrogen Life-Technologies). An automatic nuclei count was performed with the Cellomics ArrayScan VTI HCS 700 series and HCS Studio Cell Analysis Software (ThermoFisher Scientific). Data were normalized to the negative control (i.e., 0.1% DMSO) to obtain the percent of control values. The screen was performed once in triplicate. A schematic overview of the epigenetics compound screen is depicted in Figure 2A.

The efficacy of compounds that reduced cell proliferation with >60% in both cell lines was validated by repeating the screen at final concentrations of 0.1 µM, 0.5 µM, and 1 µM. The validation screen was performed once in triplicate. A schematic overview of the validation screen is depicted in Figure 2C. The online tool MORPHEUS (Broad Institute, Cambridge, MA, USA) was used to generate heatmaps.

**Figure 1 cancers-14-04708-f001:**
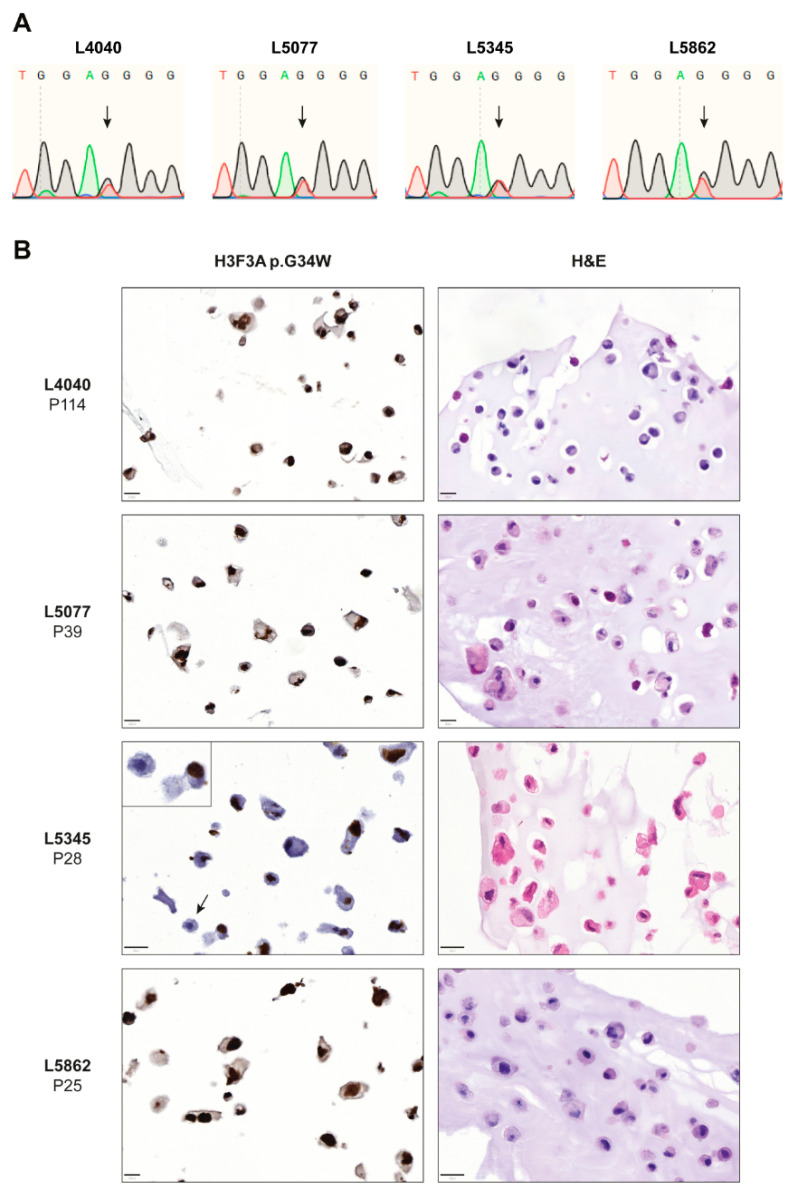
Established cell lines consist of neoplastic “stromal” cells that express the H3F3A p.G34W oncohistone in vitro. (**A**) Sanger sequencing confirmed the presence of the H3F3A p.G34W variant in all established cell lines. (**B**) H&E and H3F3A p.G34W immunohistochemical stains on paraffin-embedded cells of the four established cell lines. All cell lines showed expression of the H3F3A p.G34W mutation in culture. The L5345 cell line contained a small number of negative cells (indicated by arrow and inset). Scale bar: 20 µm.

**Figure 2 cancers-14-04708-f002:**
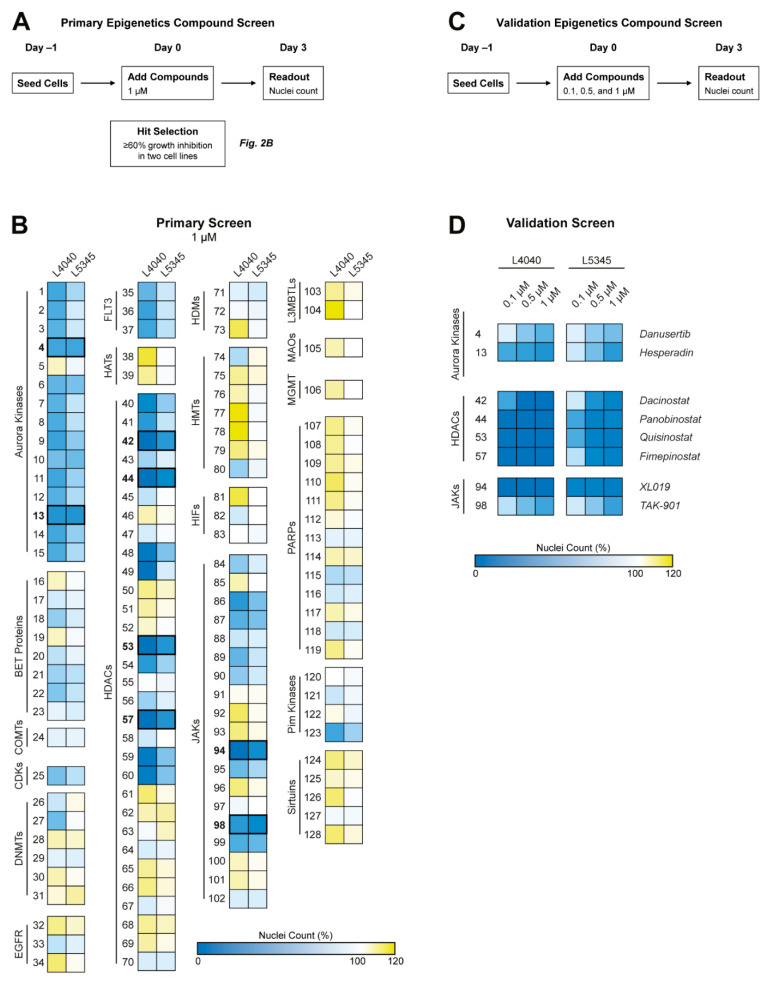
Epigenetics compound screen identified Aurora kinases, Janus kinases, and HDAC enzymes as potential therapeutic targets for GCTB. (**A**) Schematic overview of the performed primary epigenetics compound screen. (**B**) Two cell lines were treated with 128 compounds for 72 h at a concentration of 1 µM. Numbers correspond to compounds presented in Appendix A. Heatmap colors indicate growth inhibition (blue), similar growth (white), or growth induction (yellow) as compared with the 0.1% DMSO-treated control wells. Aurora kinase, HDAC, and JAK inhibitors were identified as compound classes of interest. Compounds that inhibited growth with ≥60% in both cell lines were selected for a validation screen, as indicated by the bold numbers and lines (*n* = 8). (**C**) Schematic overview of the performed validation epigenetics compound screen. (**D**) Both cell lines were treated with the eight selected compounds for 72 h at three different concentrations (i.e., 0.1, 0.5, and 1 µM). Heatmap colors are equal to (**A**). All HDAC inhibitors and one JAK inhibitor (i.e., XL019) had a pronounced effect on cellular growth at the lowest compound concentration.

### 2.6. Next-Generation RNA-Sequencing

RNA was isolated from cultured cells (i.e., L4040, L5077, and L5345) with the Direct-zol MiniPrep RNA isolation kit (Zymo Research, Irvine, CA, USA), according to the manufacturer’s protocol. RNA-sequencing on the BGISEQ-500 platform was performed by BGI Genomics (Shenzhen, China). RNA-Seq files were processed using the opensource BIOWDL RNAseq pipeline v5.0.0 (https://zenodo.org/record/5109461#.Ya2yLFPMJhE) (accessed on 15 June 2022). This pipeline performs FASTQ preprocessing (including quality control, quality trimming, and adapter clipping), RNA-Seq alignment, read quantification, and optional transcript assembly. FastQC was used for raw FASTQ read quality control. Adapter clipping was performed using Cutadapt (v2.10) with default settings. RNA-Seq reads’ alignment was performed using STAR (v2.7.5a) on the GRCh38 human reference genome. The gene read quantification was performed using HTSeq-count (v0.12.4) with the setting “–stranded=no”. The gene annotation used for quantification was Ensembl version 105. Transcripts per kilobase million (TPM) values were calculated for each sample using StringTie (v1.3.6). Variant calling was performed using GATK HaplotypeCaller (v4.1.8.0) with parameters suitable for RNAseq data analysis. Ensembl VEP was used for variant annotation. To select only meaningful variants for the follow-up analysis, we applied a filter on minimal read coverage of 10 reads (DP = 10) and minimal variant allele frequency of 30%. The analyzed data were used to determine the expression ratio of H3F3A wildtype and p.G34W (*H3-3A*) and pathogenic variants that could have caused spontaneous immortalization in L4040 cells (*ATRX*, *DAXX*, *TP53*, *RB1*, *CDKN2A*, and *CDKN2B*). Additionally, expression levels of all HDAC subtypes were extracted from the dataset. The raw data, identified variants, and obtained TPM values were deposited in the European Genome-phenome Archive (EGA) under accession number EGAS00001006441.

### 2.7. Drug Treatment of 2D Cell Cultures

L4040 (5000/well), L5077 (10,000/well), and L5862 (7500/well) cells were seeded in optimized cell densities in 96-well plates. After overnight attachment, cells were treated for 72 h with five HDAC inhibitors in concentrations ranging from 0.1 nM to 1 µM. Nuclei counts were performed as described under the methods section of the epigenetics compound screens. Data were normalized to “time 0 measurements” (i.e., nuclei counts before treatment) and the negative control (i.e., 0.1% DMSO) to correct for differences in growth rate kinetics (GR Calculator, http://www.grcalculator.org) (accessed on 7 September 2021) [23]. Experiments were performed three times in triplicate.

### 2.8. Drug Treatment of 3D Cell Cultures

Multi-cellular tumor spheroids (MCTS) of L4040 and L5862 cells (10,000/well) were generated as previously described [24] and grown for seven days before drug treatment was started. MCTS were treated with romidepsin and panobinostat in concentrations ranging from 0.1 nM to 10 µM. After 72 h of treatment, MCTS were incubated in normal growth medium containing PrestoBlue cell viability reagent (1:10 dilution) for 1.5 h at 37 °C. Fluorescence was measured at 550/600 nm with the Infinite M Plex plate reader (Tecan Group Ltd., Männedorf, Switzerland). Data were normalized to the negative control (i.e., 0.1% DMSO) to obtain the percent of control values.

Subsequently, MCTS were fixed in 4% formaldehyde containing Alcian blue (1:200 dilution) and prepared for paraffin embedding with the Epredia™ Cytoblock™ Cell Block Preparation System. H&E stains were performed using the Leica ST5020-CV5030 Stainer Integrated Workstation. The H3F3A p.G34W immunohistochemical stain (RM263, RevMAb, Biosciences) was performed with the Dako Omnis immunostainer according to the standard diagnostic procedure at LUMC. Additionally, immunohistochemistry for Ki-67 (1:1600 dilution, clone D2H10, Cell Signaling Technology (CST), Leiden, The Netherlands) and cleaved caspase 3 (Cleaved caspase 3 (Asp175), 1:800 dilution, CST) was performed as previously described [24]. Experiments were performed three times in triplicate.

### 2.9. Western Blot

L4040 (125,000/well), L5077 (300,000/well), and L5862 (225,000/well) cells were seeded in 6-well plates and allowed to attach overnight. Subsequently, cells were treated for 24 h or 48 h with romidepsin (0.1 to 3.2 nM) or panobinostat (20 to 50 nM). Sample preparation, Western blotting, and quantification were performed as previously described [25]. Western blots were examined for the expression of full-length/cleaved caspase 3 (1:1000 dilution, 8G10, CST), full-length/cleaved PARP (1:1000 dilution, 46D11, CST), and acetyl-Histone H3 Lys9 (acH3K9) (1:1000 dilution, clone C5B11, CST). As a loading control, α-Tubulin (1:30,000 dilution, DM1A, CST) was used.

### 2.10. Statistical Analysis

Outlier values in the dose–response curve datasets were identified with the online outlier calculator (Grubbs’ test, α = 0.05) from GraphPad (San Diego, CA, USA).

## 3. Results

### 3.1. Established Cell Lines Consist of Neoplastic “Stromal” Cells That Express the H3F3A p.G34W Oncohistone In Vitro

Four cell lines of neoplastic cells from GCTB patients that were pre-operatively treated with denosumab were established and all primary tumors harbored an H3F3A p.G34W mutation (Appendix A). STR profiles from the primary tumors were identical to those from the established cell lines (Appendix A). Sanger sequencing confirmed the presence of a heterozygous H3F3A p.G34W mutation in all cell lines (Figure 1A); transcriptome sequencing showed RNA expression of both the H3F3A wildtype and mutant allele in three cell lines (Appendix A). All established cell lines expressed the H3F3A p.G34W protein at passage 25 or higher as determined by immunohistochemistry with an H3F3A p.G34W mutant-specific antibody (Figure 1B), confirming outgrowth of neoplastic cells and expression of the oncohistone in vitro. Notably, the L5345 cell line contained a small number of H3F3A p.G34W-negative cells, as indicated by the arrow and inset in Figure 1B. STR profiling did not indicate contamination with another cell line in culture (Appendix A), suggesting that a non-neoplastic cell type from the primary GCTB tumor survived until passage 28 or that expression of the oncohistone was lost in a subset of cells due to a currently unknown mechanism. Transcriptome sequencing of L5345 at passage 28 showed 51% wildtype and 49% mutant *H3F3A* reads (Appendix A), confirming the presence and overall balanced expression of the H3F3A p.G34W mutation. Interestingly, the L4040 cell line reached passage numbers > 100, indicating spontaneous immortalization, although no *ATRX*, *DAXX*, *TP53*, *RB1*, *CDKN2A*, and *CDKN2B* variants were identified in the RNA sequencing data that could explain this unlimited lifespan. Hence, four novel cell lines of neoplastic cells from GCTB were successfully established. These four cell lines were subsequently used to identify novel therapeutic strategies, although not all cell lines could be included in each performed experiment due to time of establishment or slow growth rates (Appendix A).

### 3.2. Epigenetics Compound Screen Identifies Aurora Kinases, Janus Kinases, and HDAC Enzymes as Potential Therapeutic Targets for GCTB

To explore the potential of targeting the epigenetic landscape in GCTB, a broad epigenetics compound screen (*n* = 128) was performed in two cell lines to identify novel therapeutic targets in an unbiased way (Figure 2A). Several compound classes, including Aurora kinase, Janus kinase (JAK), and HDAC inhibitors, reduced the growth of both GCTB cell lines (Figure 2B).

Eight compounds which inhibited cell growth with ≥60 % in two cell lines, as indicated by the bold numbers and lines in Figure 2B, were selected for the validation screen (Figure 2C). Notably, several HDAC inhibitors and one JAK inhibitor (i.e., XL019) had a pronounced effect on cellular growth at a tenfold lower compound concentration than used in the primary screen (100 nM vs. 1 µM) (Figure 2D). Hence, Aurora kinase, JAK, and HDAC inhibitors represent promising therapeutic strategies for GCTB.

### 3.3. Class I HDAC Subtypes Are Most Abundantly Expressed in GCTB

HDAC inhibitors had the highest overall growth-reducing effect on the neoplastic cells in GCTB (average of 90% growth inhibition at 100 nM); therefore, this compound class was selected for further studies. HDAC enzymes consist of eleven different subtypes, which can be subdivided into four classes (class I, IIA, IIB, and IV). RNA expression levels of all HDACs were extracted from a previously generated RNA-sequencing dataset of three of the established cell lines (L4040, L5077, and L5345). All HDAC subtypes, except for HDAC4, were expressed in the neoplastic cells, of which class I (i.e., HDAC1, 2, and 3) and HDAC7 showed the highest levels of RNA expression (Figure 3). Hence, the targets of HDAC inhibitors are expressed in GCTB, and broad or class-I-specific HDAC inhibitors may have the most pronounced effect on the growth of the neoplastic cells.

### 3.4. HDAC Inhibitors Reduce the Growth of Neoplastic “Stromal” Cells in Both 2D and 3D In Vitro Models

The four HDAC inhibitors identified in the epigenetics compound screen target all HDAC subtypes (dacinostat, panobinostat, and quisinostat), or specifically, class I, IIB, and IV (fimepinostat). The transcriptome sequencing data indicated that class I HDAC subtypes were most abundantly expressed in the neoplastic cells; therefore, the clinically approved class-I-specific HDAC inhibitor romidepsin was also included in follow-up studies, although this specific HDAC inhibitor was not identified as a hit compound in the primary epigenetics compound screen (compound #52). These five selected HDAC inhibitors were used to derive dose–response curves in three of the GCTB cell lines, of which two were not used in the epigenetics compound screen (L5077 and L5862). All cell lines were highly sensitive to the five HDAC inhibitors (Figure 4A), with GR_50_ and IC_50_ values in the nanomolar range after 72 h of treatment (Table 1). Interestingly, all cell lines were most sensitive to treatment with romidepsin (Table 1), implying that class I HDACs are indeed the subtypes of interest in GCTB. The two clinically approved HDAC inhibitors (i.e., romidepsin and panobinostat) were selected for further studies, to improve the translation of our preclinical findings to clinical trials.

The reduced complexity of 2D cell cultures can affect drug response, and thereby impede clinical translation; thus, we established 3D in vitro models of two GCTB cell lines to validate the effect of romidepsin and panobinostat. Dose–response curves of these two HDAC inhibitors in MCTS models of L4040 and L5862 cells showed similar IC_50_ values as compared with the 2D cell culture experiments (Figure 5A and Table 1). Hence, both pan-HDAC and class-I-specific HDAC inhibitors reduce the growth of the neoplastic cells in 2D and 3D in vitro models.

### 3.5. Both Pan-HDAC and Class-I-Specific HDAC Inhibitors Induce Apoptosis and Histone Acetylation in GCTB

The inhibition of HDAC enzymes should prevent histone deacetylation, and thus cause an overall increase in histone acetylation; therefore, the levels of acetylated histone 3 (acH3K9) were determined after 24 h of treatment with romidepsin and panobinostat at ≥ GR_100_ concentrations (Figure 4A). Indeed, treatment with both HDAC inhibitors induced an overall increase in histone acetylation in the three cell lines (Figure 4B). Moreover, the effect of romidepsin on histone acetylation was dose-dependent (Figure 4C), suggesting that lower HDAC inhibitor concentrations already affect the epigenetic landscape of the neoplastic cells, whereas growth is still minimally affected (Figure 4A).

To assess the underlying cell death mechanism, three cell lines were treated for 48 h with romidepsin and panobinostat at ≥GR_100_ concentrations (Figure 4A). Romidepsin and panobinostat both increased the cleavage of PARP and caspase 3, both indicators of ongoing apoptosis, in all three cell lines, although caspase 3 activation was more pronounced than PARP cleavage (Figure 4D). Apoptosis induction was also observed in the 3D in vitro models of L4040 and L5862 after 72 h treatment with 10 nM romidepsin and 100 nM panobinostat (~IC_80_ values in 3D in vitro models). A reduction in proliferation was observed in the L4040 cell line, whereas L5862 cells showed a slight increase in proliferation after treatment (Figure 5B). Notably, some L4040 and L5862 cells lost the detectable expression (IHC) of the oncohistone in untreated as well as treated conditions, suggesting that unknown factors can influence the expression levels of H3F3A p.G34W. Together, these results indicate that both pan-HDAC and class-I-specific HDAC inhibitors induce apoptosis in and alter the epigenetic landscape of the neoplastic cells.

## 4. Discussion

In this study, four novel cell lines of the neoplastic “stromal” cell component of GCTB were established and used to examine whether targeting the epigenetic landscape could be a promising therapeutic strategy to eliminate the neoplastic cells from GCTB. The establishment of novel patient-derived cell lines of neoplastic “stromal” cells is a great asset to the field, because the limited amount of available in vitro models restricts research on rare tumor entities such as GCTB [26]. To the best of our knowledge, we are the first to report on a GCTB cell line with an unlimited lifespan (L4040), which was not actively immortalized and does not show mutations in *ATRX*, *DAXX*, *TP53*, *RB1*, *CDKN2A*, and *CDKN2B*. Previous studies showed a maximum passage number of 20 to 40 [27,28,29], which is in line with the other three cell lines that were established in this study (L5077, L5345, and L5862). All established cell lines consist of neoplastic “stromal” cells that express the H3F3A p.G34W oncohistone whilst culturing over time.

In general, compound screens are a rough and easy way to explore drug sensitivity on a large scale, although have limitations, thus always requiring validation of “hits”. Pitfalls may include pipetting mistakes or inactivity of compounds, leading to false-negative or false-positive results. For instance, romidepsin (compound #52) was not identified as a hit compound in the primary epigenetics compound screen, although further studies showed the efficacy of romidepsin in three GCTB cell lines, which is a false-negative result that we cannot explain. Therefore, the current study only focused on compounds that showed an effect in the epigenetics compound validation screen.

We identified three compound classes that affect the cellular growth of the neoplastic cells: Aurora kinase, JAK, and HDAC inhibitors. TAK-901 is officially classified as an Aurora kinase inhibitor, but has a tenfold higher affinity for JAK3, making it a potent dual inhibitor of two interesting compound classes. Of interest, the most effective JAK inhibitor (i.e., XL019) also targets Platelet-Derived Growth Factor Receptor Beta (PDGFRβ), which is a receptor tyrosine kinase that is highly phosphorylated in GCTB [30]. Inhibition of PDGFRβ signaling with sunitinib affects the cell viability of the neoplastic cells in vitro and caused the complete depletion of giant cells and neoplastic cells in one patient [30]. Hence, compounds that target Aurora kinases, JAKs, and PDGFRβ warrant further investigation to determine whether these targeted therapies are of interest to treat GCTB.

The most promising therapeutic strategy identified in the current study was the use of HDAC inhibitors, which are a group of enzymes that play an important role in normal bone and cartilage development. HDAC1, -3, -5, and -7 are most abundantly expressed in bone tissue [31], and our study showed that three of these HDAC subtypes (HDAC1, -3, and -7) are also most prominently expressed in the GCTB cell lines. However, the overall expression patterns of HDAC enzymes in the neoplastic cells do not seem to resemble that of normal bone tissue [31]. The aberrant expression of HDAC enzymes can cause several bone-related abnormalities, including increased RANK ligand expression and bone resorption caused by the depletion of HDAC4 and -5 [32]. Notably, the expression of HDAC4 is indeed lost in the neoplastic cells. To elucidate whether HDAC enzymes contribute to the formation of GCTB and how HDAC inhibitors might be able to counteract the aberrant HDAC expression levels, additional studies should be performed.

Recently, it was also shown by others that romidepsin affects cell viability in four other patient-derived GCTB cell lines [28,29]. Moreover, it was shown that givinostat, a class I, IIA, and IIB HDAC inhibitor, reduced tumor growth in a patient-derived xenograft mouse model of malignant GCTB [33]. Interestingly, givinostat (compound #48) also showed an effect in the primary epigenetics compound screen that was performed in the current study, although the effect was not pronounced enough to be selected as a hit compound (95% and 44% growth reduction in L4040 and L5345 cells, respectively). Nevertheless, we showed that the neoplastic cells are highly sensitive to several broad HDAC inhibitors (i.e., dacinostat, quisinostat, panobinostat, and fimepinostat), as well as an HDAC class-I-specific inhibitor (i.e., romidepsin). Together, the current study and the studies mentioned above suggest that HDAC inhibition could be a potential therapeutic strategy for GCTB patients, although additional research is needed to elucidate the underlying biological mechanism for HDAC inhibitor sensitivity in GCTB.

Panobinostat and romidepsin are clinically approved for the treatment of multiple myeloma and peripheral T-cell lymphomas, respectively, and reach maximum plasma concentrations of 62 nM and 700 nM, respectively [34,35]. The maximum plasma concentrations in patients are comparable to the obtained GR_50_ and IC_50_ values for both HDAC inhibitors in our established 2D and 3D in vitro models of GCTB, suggesting that our findings are not related to off-target or toxic side-effects. However, HDAC inhibitors can cause serious adverse events in patients, including thrombocytopenia, neutropenia, various cardiac and metabolic effects, and infections [36]. The side-effects of long-term denosumab are limited; therefore, it should be carefully assessed per individual GCTB patient which type of treatment is most suitable and whether the elimination of neoplastic cells by HDAC inhibition outweighs the risk of potential side-effects.

Although this study clearly shows that HDAC inhibitors can eliminate the neoplastic cells in GCTB, the study design has limitations in regard to finding novel therapeutic options for GCTB. First, the utilized in vitro models do not entail all cell types that are normally present in GCTB lesions. The interplay between the neoplastic cells, monocytes, and giant cells might be an important characteristic to consider when identifying therapeutic strategies for GCTB patients. Additional research in co-culture 3D in vitro models of neoplastic cells and monocytes, as well as orthotopic in vivo models of GCTB, should be performed to confirm our current findings before clinical trials can be initiated. Moreover, these models could be used to investigate whether reactive cell reducing treatment (i.e., denosumab) would enhance the efficacy of HDAC inhibitors, which could potentially lead to reducing the dose of HDAC inhibitors to avoid side-effects in patients. The current study shows that at least one HDAC inhibitor affected the epigenome of neoplastic cells at concentrations which minimally affected cell viability, indicating the potential of combination strategies with HDAC inhibitors. Second, HDAC inhibitor sensitivity could not be directly linked to the presence of the H3F3A p.G34W mutation. It would be of interest to establish isogenic cell line pairs to study whether a synthetic lethal interaction between HDAC inhibitors and the H3F3A p.G34W mutation exists and whether treatment with this compound class can reverse H3F3A p.G34W-induced remodeling of the epigenome, which may also elucidate the underlying biological mechanism of HDAC inhibitor efficacy in GCTB. Finally, the current drug screen was focused on epigenetic targets and monotherapies, although targets beyond the epigenome as well as combination strategies are also of interest for GCTB. Recent studies indicate that next-generation antiangiogenic drugs could be promising therapeutic strategies for GCTB. Vascular endothelial growth factor (VEGF) expression is associated with the aggressiveness of osteolytic bone lesions [37] and treatment with lenvatinib with or without denosumab reduces the viability of primary GCTB cultures [38]. Lenvatinib combined with chemotherapeutic agents shows promising antitumor activity in osteosarcoma patients [39], and a combination strategy with pembrolizumab is currently under evaluation in a wide range of sarcoma patients, including bone sarcoma patients (NCT04784247).

## 5. Conclusions

In summary, we established four novel cell lines of neoplastic “stromal” cells of GCTB which can be used to identify novel therapeutic targets as well as to better understand tumor biology in GCTB. Our study identified HDAC inhibitors as a potential therapeutic option to eliminate the neoplastic cells from GCTB lesions, although further studies should be performed to elucidate the mechanisms underlying HDAC inhibitor sensitivity in GCTB. Nevertheless, the identification of such therapeutic strategies is an important step towards the treatment of patients with unresectable or recurrent tumors who currently rely on life-long treatment with denosumab.

## Figures and Tables

**Figure 3 cancers-14-04708-f003:**
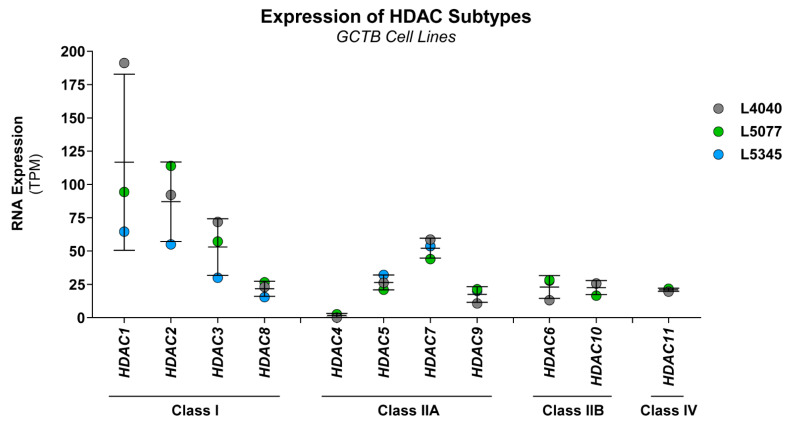
Class I HDAC subtypes are most abundantly expressed in GCTB. RNA expression levels of all eleven HDAC subtypes in L4040 (p100), L5077 (p28), and L5345 (p27) cells. Data were extracted from a previously generated RNA-sequencing dataset. Gene expression levels are presented as transcripts per kilobase million (TPM) values. Data points represent the TPM value for each cell line, and mean ± standard deviation was calculated. All HDAC subtypes were expressed in the cultured cells, except for HDAC4. HDAC1, 2, 3, and 7 were most abundantly expressed in all cell lines.

**Figure 4 cancers-14-04708-f004:**
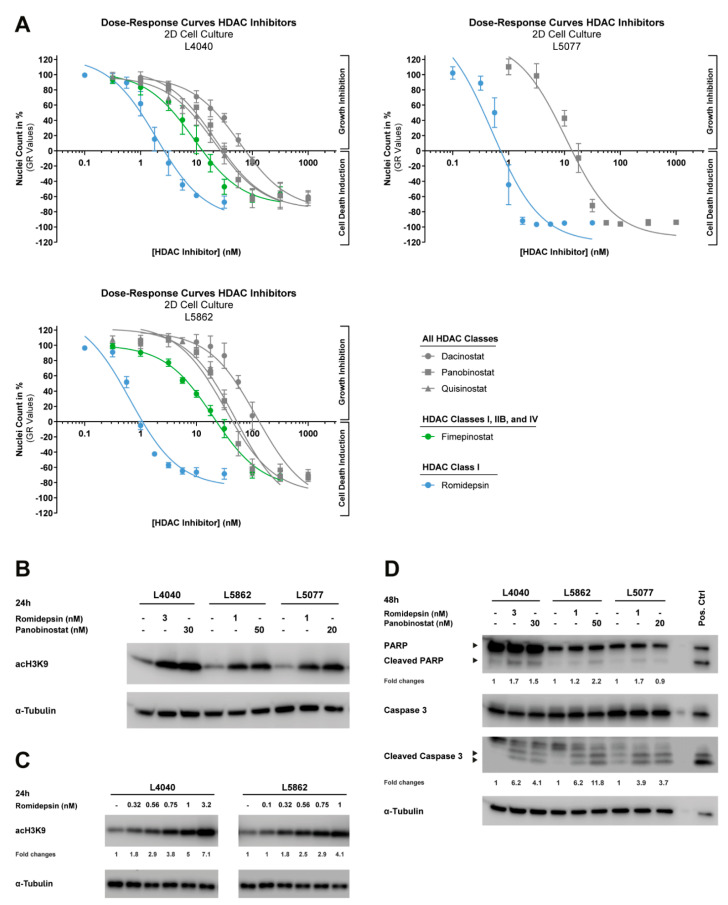
HDAC inhibitors reduce growth and induce apoptosis and histone acetylation in 2D in vitro models of neoplastic “stromal” cells. (**A**) Dose–response curves of five HDAC inhibitors (i.e., dacinostat, panobinostat, quisinostat, fimepinostat, and romidepsin) after 72 h of treatment for three cell lines cultured in 2D. Data were corrected for differences in growth rates between cell lines (GR values). Data points represent the means of three experiments performed in triplicate ± standard deviation. All cell lines are highly sensitive to HDAC inhibitor treatment and showed GR_50_ values in the nanomolar range (Table 1). (**B**) Western blot for acH3K9 after 24 h treatment with 1 to 3 nM romidepsin and 20 to 50 nM panobinostat. α-Tubulin was used as a loading control. Original blots are depicted in Appendix A. Histone acetylation was induced in three cell lines after treatment with romidepsin or panobinostat. (**C**) Western blot for acH3K9 after 24 h treatment with increasing doses of romidepsin (0.1 to 3.2 nM). α-Tubulin was used as a loading control. Fold changes are relative to α-Tubulin expression and untreated controls. Original blots are depicted in Appendix A. Two cell lines showed a dose-dependent increase in histone acetylation after treatment with nanomolar concentrations of romidepsin. (**D**) Western blot for PARP and caspase 3 (cleaved and full-length) after 48 h of treatment with 1 to 3 nM romidepsin and 20 to 50 nM panobinostat. CH2879 chondrosarcoma cells treated for 24 h with 5 µM ABT-737 + 1 µM doxorubicin were used as a positive control. α-Tubulin was used as a loading control. Fold changes are relative to α-Tubulin expression and untreated controls. Original blots are depicted in Appendix A. Treatment with both HDAC inhibitors induced apoptosis in three cell lines.

**Figure 5 cancers-14-04708-f005:**
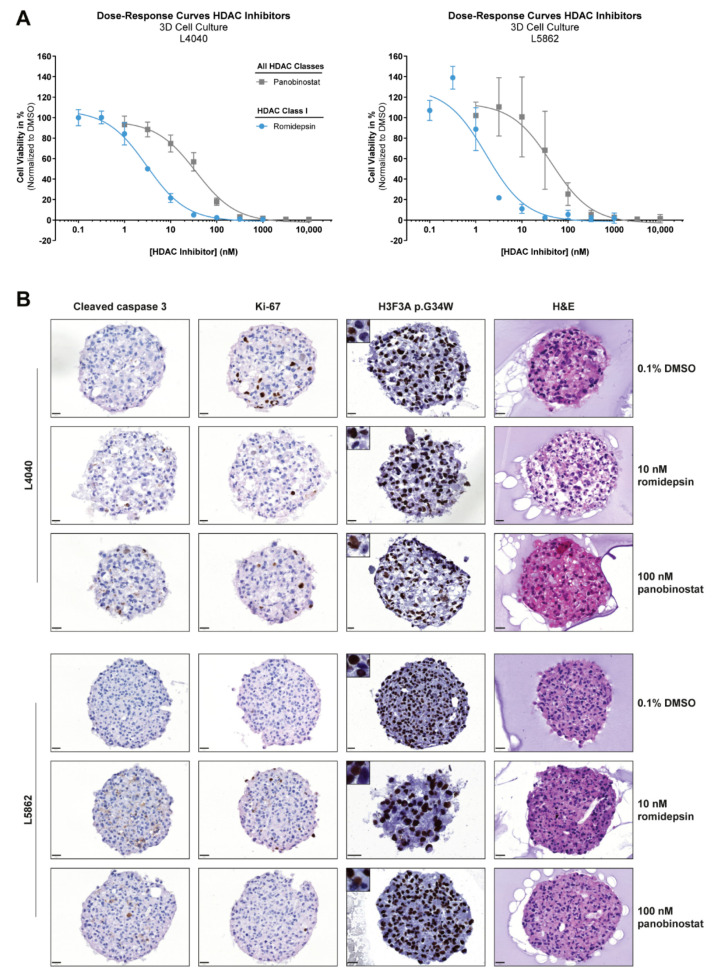
HDAC inhibitors reduce growth and induce apoptosis in 3D in vitro models of neoplastic “stromal” cells. (**A**) Dose–response curves of two HDAC inhibitors (i.e., panobinostat and romidepsin) after 72 h of treatment for cell lines cultured in 3D. Data points represent the mean of three experiments performed in triplicate ± standard deviation. HDAC inhibitor sensitivity was comparable between 2D and 3D in vitro models (Table 1). (**B**) H&E, H3F3A p.G34W, Ki-67, and cleaved caspase 3 stains on sections of L4040 and L5862 MCTS after 72 h treatment with 0.1% DMSO, 10 nM romidepsin, and 100 nM panobinostat. Images of single MCTS are representative of technical and biological triplicates. Both HDAC inhibitors induced apoptosis (increased cleaved caspase 3 levels) in L4040 and L5862 MCTS as compared with the 0.1% DMSO-treated controls. A reduction in proliferation (reduced Ki-67 levels) after HDAC inhibitor treatment was only observed in L4040 MCTS. Scale bar: 20 µm.

**Table 1 cancers-14-04708-t001:** GR_50_ and IC_50_ values after 72 h treatment with HDAC inhibitors in GCTB cell lines.

		Dacinostat		Panobinostat		Quisinostat		Fimepinostat		Romidepsin	
		All Classes		All Classes		All Classes		Class I, IIB, and IV		Class I	
Cell Line	In Vitro Model	GR_50_ (nM)	IC_50_ (nM)	GR_50_ (nM)	IC_50_ (nM)	GR_50_ (nM)	IC_50_ (nM)	GR_50_ (nM)	IC_50_ (nM)	GR_50_ (nM)	IC_50_ (nM)
L4040	2D	23.2	24.8	12.5	12.7	8.45	9.37	4.88	5.31	1.25	1.34
	3D	-	-	-	35.3	-	-	-	-	-	3.02
L5077	2D	-	-	9.41	18.3	-	-	-	-	0.58	0.87
L5862	2D	61.0	78.0	23.7	29.5	25.3	30.4	7.44	11.3	0.59	0.70
	3D	-	-	-	43.7	-	-	-	-	-	1.69

## Data Availability

The manuscript and Appendix A contain all potential findings based on raw data analysis. The RNA sequencing dataset was deposited in the European Genome-phenome Archive (EGA) under accession number EGAS00001006441. Other raw data are available from the corresponding author on reasonable request.

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
