# Peer review of "Histone Deacetylase Inhibitors as a Therapeutic Strategy to Eliminate Neoplastic “Stromal” Cells from Giant Cell Tumors of Bone"

_cancers, 2022, doi:10.3390/cancers14194708_

Round 1
Reviewer 1 Report
Dear authors,
Thank you for sending your manuscript titled
“Histone deacetylase inhibitors as a therapeutic strategy to eliminate neoplastic “stromal” cells from giant cell tumors of bone"
It was a pleasure to review this exciting paper aimed at finding an alternative way to treat GCTB.
Since the first attempts to identify the pathology of giant cell tumours undertaken by Nelaton in the nineteenth century through the works of J.C Bloodgood, who proposed the term Giant Cell Tumour in 1912, it is still one of the most imperfectly understood neoplasms of bone.
Although surgical treatment has not changed in decades and at present is not amenable to any significant updates, it is of paramount interest to every oncologist or orthopaedic surgeon to find new options for conservative treatment of GCTB.
Due to its locally aggressive nature and extremely high recurrence potential, GCTB can be debilitating for patients causing severe functional disability in relatively young and otherwise fit adults.
This is, therefore of the highest importance to promote articles that are finding alternative ways, primarily through targeted therapies and molecular pathways.
Using relatively new class of anti-cancer agents like histone deacetylase (HDAC) inhibitors is very promising as popular systemic treatment of GCTB with denosumab injections for locally advanced or inoperable tumours does not target neoplastic cells which are responsible for recurrence of the tumour.
There is an urgent need to develop novel therapies targeting neoplastic cells in GCTB. After discontinuing the denosumab for inoperable cases, we usually observe rapid local relapses caused by the above cells.
In their paper, authors have directly examined novel therapies using cultured cell lines obtained from resected GCTB that underwent preoperative denosumab injections. These multi-cellular tumour spheroids were treated with romidepsin and panobinostat; both histone deacetylase inhibitors used to treat multiple myeloma and peripheral T-cell lymphomas. All primary tumours treated with denosumab and harvested for cell lines harboured an H3F3A p.G34W mutation.
The results of the study are promising as HDAC inhibitors show the potential to eliminate neoplastic cells from GCTB. This can be a promising solution for patients that rely on life-long or rechallenge treatment with denosumab for inoperable tumours or even for those with locally advanced GCTB that are subject of curettage.
It is, however, essential to weigh the risk of the proposed HDAC treatment versus denosumab treatment, which, even on a long-term basis, have relatively transient and insignificant side effects.
Once again, I would like to congratulate the authors for this interesting article, and I recommend this as quite innovative and vital in the battle with GCTB.
Reviewer 2 Report
The authors report about the establishment of 4 different giant cell tumors of bone (GCTB) cell lines in which expression of H3F3A mutation on histone H3.3 was assessed through Sanger sequencing. Interestingly, one of these cultures exceeded 100 passages likely showing unlimited lifespan. The study examined the effect of targeting epigenome in eliminating neoplastic cells from GCTB. 128 compounds were screened and HDAC inhibitors were identified as most promising class. Therefore, 5 HDAC inhibitors were tested both in 2D and 3D spheroid GCTB cultures. All the cell lines were sensitive to HDAC inhibitors and also showed increased histone acetylation and apoptosis.
The manuscript is of interest since it addresses the unmet clinical need of an effective drug treatment option for unresectable GCTB or in neoadjuvant setting. The results are clearly presented and conclusions are supported by robust data. Methods are appropriate and pictures are of high quality.
However, some minor issues should be addressed:
1)In the Introduction section the authors highlight the need to develop novel therapeutic strategies which directly target the neoplastic cells. This is entirely true, and in this regard recent findings highlight the role of some TKI inhibitors, in particular the anti-VEGFR lenvatinib, in the treatment of GCTB, either alone or in combination with other agents. Indeed, it is known that cytokines such as VEGF exert a relevant role in tumor development and progression. In a recent study the administration of lenvatinib showed a higher efficacy in reducing survival of GCTB primary cultures compared to denosumab. Moreover, efficacy of lenvatinib is currently under evaluation also in the treatment of some other bone lesions such as osteosarcoma and chondrosarcoma (NCT04784247) in combination with pembrolizumab or ifosfamide/etoposide. Altogether, these data strongly highlight the promising role of new generation antiangiogenic drugs as a novel therapeutic strategy in the management of GCTB.
In this regard, it's suggested to add few sentences to discuss this topic including the following references:
- De Vita, A et al: “A Rationale for the Activity of Bone Target Therapy and Tyrosine Kinase Inhibitor Combination in Giant Cell Tumor of Bone and Desmoplastic Fibroma: Translational Evidences.” Biomedicines vol. 10,2 372. 3 Feb. 2022
- Kumta, S et al: "Expression of VEGF and MMP-9 in giant cell tumor of bone and other osteolytic lesions." Life Sci. 2003, 73, 1427–1436.
- Gaspar, N. et al. "Lenvatinib with etoposide plus ifosfamide in patients with refractory or relapsed osteosarcoma (ITCC-050): A multicentre, open-label, multicohort, phase 1/2 study." Lancet Oncol. 2021, 22, 1312–1321
2) Study limitations should be stressed
Reviewer 3 Report
cancers-1924096, Histone deacetylase inhibitors as a therapeutic strategy to eliminate neoplastic “stromal” cells from giant cell tumors of bone
The manuscript submitted for publication has a very good quality. It is well designed and the research performed seems to be correct in terms of both data acquisition and data analysis. I didn’t find anything significant that should be corrected.
